# Association between homicide rates and suicide rates: a countrywide longitudinal analysis of 5507 Brazilian municipalities

Daiane Borges Machado,[1,2] Keltie McDonald ,[3] Luis F S Castro-de-Araujo,[1,4] Delan Devakumar,[5] Flávia Jôse Oliveira Alves,[1] Lígia Kiss,[5] Glyn Lewis,[3] Mauricio L Barreto[6]

DBM and KM contributed equally.

DBM and KM are joint first authors.

For numbered affiliations see end of article.

**Correspondence to**
Dr Keltie McDonald;
k.mcdonald@ucl.ac.uk

## ABSTRACT

**Objective** To estimate the association between homicide and suicide rates in Brazilian municipalities over a period of 7 years.

**Design** We conducted a longitudinal ecological study using annual mortality data from 5507 Brazilian municipalities between 2008 and 2014. Multivariable negative binomial regression models were used to examine the relationship between homicide and suicide rates. Robustness of results was explored using sensitivity analyses to examine the influence of data quality, population size, age and sex on the relationship between homicide and suicide rates.

**Setting** A nationwide study of municipality-level data.

**Participants** Mortality data and corresponding population estimates for municipal populations aged 10 years and older.

**Primary and secondary outcome measures** Age-standardised suicide rates per 100 000.

**Results** Municipal suicide rates were positively associated with municipal homicide rates; after adjusting for socioeconomic and demographic factors, a doubling of the homicide rate was associated with 22% increase in suicide rate (rate ratio=1.22, 95% CI: 1.13 to 1.33). A dose–response effect was observed with 4% increase in suicide rates at the third quintile, 9% at the fourth quintile and 12% at the highest quintile of homicide rates compared with the lowest quintile. The observed effect estimates were robust to sensitivity analyses.

**Conclusions** Municipalities with higher homicide rates have higher suicide rates and the relationship between homicide and suicide rates in Brazil exists independently of many sociodemographic and socioeconomic factors. Our results are in line with the hypothesis that changes in homicide rates lead to changes in suicide rates, although a causal association cannot be established from this study. Suicide and homicide rates have increased in Brazil despite increased community mental health support and incarceration, respectively; therefore, new avenues for intervention are needed. The identification of a positive relationship between homicide and suicide rates suggests that population-based

## Strengths and limitations of this study

► In this longitudinal ecological study, we examined the association between homicide and suicide rates using data from 5507 municipalities in Brazil from 2008 to 2014, including several sensitivity analyses to address limitations of previous examinations on this topic.

► Homicide and suicide are important public health concerns in Brazil and globally; our findings provide clear evidence for their relationship and suggest potential avenues for public health and policy interventions to reduce deaths from these causes.

► Our study used an ecological design, and therefore, results are generalisable to the municipal population-level and not to individuals.

interventions to reduce homicide rates may also reduce suicide rates in Brazil.

## INTRODUCTION

Suicide is an important global public health problem, causing over 800 000 deaths per year. Among 15–29 year-olds, suicide is the second leading cause of mortality worldwide.[1] One of the strongest and most consistently identified risk factors for suicide is psychiatric disorder.[2] In Brazil in 2012, the national rate of suicide was 6.2 per 100 000 inhabitants and was higher in males (10.0 per 100 000) than females (2.7 per 100 000).[3] The rate of suicide has increased in nearly all age groups for males and females between 2000 and 2016.[4]

In 2008, the WHO reported violence as a leading cause of death and disability and a 'global challenge'.[1] Violence is not exclusively a criminal justice issue; it is also a health and human rights issue. Over 90% of deaths caused by violence occur in low-income and middle-income countries (LMICs); however, most studies into violent deaths have been

carried out in high-income countries (HICs).[5] In Brazil, the homicide rate has increased over the last 3 decades.[6] In 2016, interpersonal violence was the second leading cause of age-standardised years of life lost in Brazil.[7]

Brazil is the largest country in Latin America with 207 million inhabitants. Despite being the ninth richest country in the world in terms of nominal gross domestic product, Brazil has one of the highest levels of income inequality in the world.[8 9] Homicide in Brazil has been related to socioeconomic inequalities,[6 10 11] poverty,[12] race, gender and age, with men, younger people and those identifying as black at higher risk,[6 10] while suicide has also been associated with income inequality, sex, and age and low levels of education.[3 4]

While often studied separately, there is a long tradition of viewing homicide and suicide as related phenomena.[13] Homicide and suicide can be viewed similarly as forms of violence.[1] Several studies have examined the relationship between homicide and suicide rates in the population; however, evidence is mixed. There appears to be strong variations in the magnitude and direction of association across countries and regions.[14 15] Many of the previous studies of the association have been conducted in HICs. However, the relationship between homicide and suicide rates in LMICs might be different due to different distributions and influences of risk factors in the population. There has been only one previous study of the ecological association between homicide and suicide rates in Brazil, which observed an inverse correlation in 2010.[16] However, this study used data from a single year, studied large, heterogeneous geographical areas (27 Brazilian states) and did not adjust for important demographic and socioeconomic factors. We aimed to further investigate the possible association between homicide and suicide rates in Brazil, using a dataset of all Brazilian municipalities from 2008 to 2014.

## METHODS
### Study design
This study used a longitudinal ecological design, using yearly data from the 5507 Brazilian municipalities for the period 2008–2014.

### Data sources
All data used in this study are publicly available online (http://www2.datasus.gov.br). To estimate suicide and homicide rates, cause-specific mortality data were collected from the Brazilian Ministry of Health's Mortality Information System. All deaths are recorded in this system using the International Classification of Diseases, 10th revision.[17] Socioeconomic, demographic variables (monthly per capita income in Brazilian Real (BR$); unemployment rate; percentage of individuals who were divorced; percentage of individuals who declared being Pentecostal; percentage of households with only one resident; and urbanisation) and municipality population estimates were obtained from the Brazilian Institute of

Geography and Statistics (IBGE). Data from the main mental health services in Brazil, the Psychosocial Care Centre (CAPS) coverage was obtained from the Mental Health Coordination Database, held by the Ministry of Health (DATASUS).

### Definitions of variables
#### Outcome
Suicide is defined as a death resulting from intentional self-harm according to the International Classification of Diseases, 10th revision (ICD-10) codes X60 to X84.[17] In Brazil, all deaths due to external causes (suicide, homicide and accidents) are forwarded to the Medical Legal Institute (IML; artigo 2° da Resolução CFM n°. 1.779/2005) where the death certificates are printed and signed by an examining doctor,[18] which decreases the chances for misclassification. Diagnoses are based on an autopsy (using the ICD-10 classification), to investigate the circumstances in which the death occurred, personal history of the victim and suicide risk factors.[19]

The suicide rate was calculated at the municipal level and as it is influenced by age, we standardised by each 5-year age range, from 10 years or older. We used direct standardisation using the WHO population distribution as the standard population. Suicides among individuals under 10 years old were excluded because suicide is a very rare event before 10 years; there were 33 victims of suicide under 10 years in Brazil in the period 2008–2014 . Overall and sex-specific age-standardised suicide rates were calculated for each municipality and year of analysis expressed as a rate per 100 000 inhabitants.

#### Exposure
Homicide was defined as a death resulting from injuries inflicted by another person with the intent to injure or kill using any means according to the ICD-10 codes X85 to Y09.[17] Homicide rates were also calculated at the municipal level. Homicide rates were not standardised by age because it was used as a proxy for violence, and therefore, total violence irrespective of age distribution was of interest. Overall and sex-specific homicide rates were calculated for each municipality and year of analysis expressed as a rate per 100 000 inhabitants.

#### Confounders
Municipal-level covariates for suicide considered in the model included: the monthly per capita income in Brazilian Real (BR$); unemployment rate, measured by the percentage of the population aged 16 years or over with no official job; percentage of individuals who were divorced; percentage of individuals who declared being Pentecostal; percentage of households with only one resident; and urbanisation, measured as the percentage of individuals living in urban areas for each municipality. These variables were selected based on evidence in the literature that they may be associated with suicide.[3 20–22]

The percentage of individuals who declared being Pentecostal was chosen as a confounder, as in previous

studies,[3 22] because it is likely to be the best proxy for measuring religious engagement in Brazil; Brazilians who identify themselves as Pentecostal tend to actively attend the church and its activities, while those who identify themselves as Catholic seem to do so only referring to their family background, even if they do not participate in any religious activity.[23]

We also included the CAPS coverage, defined as the proportion of people who could potentially be seen in the centres, because availability of mental healthcare may be associated with suicide rates. CAPS coverage was calculated for each municipality based on the different types of CAPS available and the populations that they are expected to serve (online supplemental methods).

## Statistical analysis

We used negative binomial regression analysis to evaluate the association between age-standardised suicide rates and rates of homicide. Year of study (a time-specific fixed effect) was introduced into the models to control for the national-level policy changes or secular trends that may have affected all municipalities.[24]

The main estimated regression model was: $\log(Y_{it}) = \alpha_i + \beta_1 HOMICIDE_{it} + \beta_n X_{nit} + \gamma_t + u_{it}$, where t=year and i=municipality, $Y_{it}$ is the age-standardised mortality rate (per 100 000 inhabitants) due to suicide, $\alpha_i$ is the fixed effect for municipality i capturing unobserved time-invariant factors causing between-municipality variability in suicide, $HOMICIDE_{it}$ was the rate of homicide for the municipality i in the year t, $X_{nit}$ is the value taken by a covariate vector containing the n socioeconomic and demographic determinants included in the model, for municipality i in year t, $\gamma_t$ is a time-specific effect, capturing variability in suicide rates over time, $u_{it}$ is an error term, assumed to follow a normal distribution. To produce rate ratio (RR) estimates that were more readily interpretable, the homicide rate was divided by 100 (ie, homicide rate per 1000).

The age-standardised suicide rate, although not a count, still appeared to follow a binomial distribution (online supplemental figure 1). We compared a negative binomial model to a Poisson regression model using an Akaike's Information Criterion and the Bayesian Information Criterion to establish the model that best fitted the data.[25] We found evidence that the age-standardised suicide rates varied between municipalities more than expected under a Poisson distribution; therefore, the negative binomial model was used to allow for overdispersion. The Hausman test was used to compare a negative binomial regression with fixed effects to a model with a random effect for each municipality. The test showed no evidence that the random effects model provided a better fit to the data (p=0.05); therefore, our model included only fixed effects.

Population standardisation was used to allow comparability across municipalities with distinct age structures. We also tested the association using models based on both age-standardised homicide rates and age-standardised suicide rates, and the results were similar (online supplemental table 1).

## Primary analysis

To test the association between homicide rates and age-standardised suicide rates, we constructed a negative binomial regression model including data from all Brazilian municipalities. We produced both unadjusted models and models adjusting for confounders.

## Secondary analyses

To examine the robustness of our primary findings, we fitted several additional negative binomial regression models: one including only municipalities considered to have very accurate vital information according to a previous study,[26] three by total municipal population (10 000 inhabitants; 10 001–50 000; and more than 50 000 inhabitants), one by municipal homicide rates quintiles and six further stratified by sex and age groups (10–19 years; 20–59 years; 60 years and older). Finally, we tested the association between homicide rates and age-standardised suicide rates for 2010 only to enable a comparison with the results of a previous study of Brazilian states.[16] For each analysis, we fitted both unadjusted and adjusted models. Statistical analyses were performed using Stata V.15 software.[27]

## Patient and public involvement

There was no patient and public involvement in this study.

## RESULTS

The age-standardised suicide rate increased in the Brazilian municipalities from 7.6/100 000 in 2008 to 8.2/100 000 inhabitants in 2014. Homicide rates also increased over this period, from 15.2/100 000 to 19.6/100 000 inhabitants. Unemployment decreased between 2008 and 2014, while all other covariates increased (table 1).

We found a positive association between homicide rates and age-standardised suicide rates in the unadjusted model and after controlling for confounders (table 2). In the model based on all Brazilian municipalities, a doubling of the homicide rate was associated with 22% increase in the suicide rate (RR=1.22, 95% CI: 1.13 to 1.33). We found similar results in the models stratified by municipal population sizes. Sensitivity analyses, based on a subset of municipalities considered to have accurate data, yield similar results but with a stronger association (RR=1.44, 95% CI: 1.22 to 1.70).

Analysing by quintile of homicide rate showed that the higher the homicide rate, the stronger the association, with 1% suicide rate increase at the second quintile, 4% increase at the third quintile, 9% increase at the fourth quintile and 12% at the highest quintile compared with those in the lowest quintile (table 3). The analysis stratified by sex and age groups showed a positive association between homicide rates and suicide among men from 20 to 59 years old (table 4).

**Table 1** Mean values and SD of suicide and homicide rates and selected covariates for the Brazilian municipalities (n=5507) in 2008 and 2014

| Outcome | 2008 | | 2014 | | Percentage change* |
|---|---|---|---|---|---|
| | Mean | SD | Mean | SD | |
| Suicide rate†‡ | 7.58 | 12.79 | 8.18 | 13.25 | 7.97 |
| Homicide rate† | 15.21 | 18.39 | 19.65 | 21.84 | 29.16 |
| CAPS coverage per municipal population | 0.26 | 0.73 | 0.45 | 0.97 | 72.13 |
| Monthly per capita income (BR$) | 453.30 | 227.58 | 540.53 | 265.35 | 19.24 |
| Unemployed people (%) | 7.15 | 3.72 | 4.85 | 4.22 | −32.16 |
| Individuals who were divorced (%) | 3.37 | 1.75 | 4.13 | 2.10 | 22.73 |
| Pentecostal Christians (%) | 10.43 | 6.17 | 12.10 | 7.15 | 15.96 |
| Households with one resident (%) | 11.02 | 2.88 | 12.82 | 3.33 | 16.27 |
| Urbanisation (%) | 63.04 | 22.03 | 66.26 | 21.80 | 5.10 |

*Percentage change from 2008.
†Rate per 100 000.
‡Age-standardised.
CAPS, Psychosocial Care Centre.

We repeated the analysis based on data from 2010 only. The unadjusted analysis showed a significant inverse relationship between homicide rates and suicide rates in 2010 (RR=0.51, 95% CI: 0.46 to 0.64). However, after adjusting for confounders, the estimate of association became closer to the null value (RR=1.0) and was no longer significant (RR=0.91, 95% CI: 0.77 to 1.08; online supplemental table 2).

## DISCUSSION
### Main findings
We assessed the relationship between homicide and suicide rates in Brazilian municipalities over a 7-year period. We found that municipalities with higher homicide rates also had higher age-standardised suicide rates in both unadjusted and adjusted models. Our results appeared to be robust based on several sensitivity analyses.

Similar results were observed for models stratified by population size and for models including only municipalities considered to have accurate data.[26] The association between homicide and suicide rates seemed to follow a dose–response pattern, such that areas with higher homicide rates showed stronger relationships with increased suicide rates, reaching 12% increase in the highest quintile of homicide rates compared with the lowest.

### Strengths and limitations
This study examined the relationship between homicide and suicide rates in detail. Our data covered all 5507 municipalities in Brazil and are, therefore, representative of the country. The use of fine-grained geographical areas likely resulted in less within-area heterogeneity and provided improved opportunity to control for confounding factors. The estimates of association observed in this study appear

**Table 2** Unadjusted and adjusted associations between suicide rates and homicide rates in all the Brazilian municipalities, only municipalities with accurate vital information in Brazil, and by population size, 2008–2014

| | Unadjusted model | | Adjusted model* | | Number of suicides | Number of municipalities‡ |
|---|---|---|---|---|---|---|
| | RR† | 95% CI | RR† | 95% CI | | |
| All Brazilian municipalities | 1.25 | 1.16 to 1.36 | 1.22 | 1.13 to 1.33 | 35 357 | 5051 |
| Municipalities with accurate vital information | 1.45 | 1.23 to 1.70 | 1.44 | 1.22 to 1.70 | 9336 | 1556 |
| Population ≤10 000 | 1.19 | 1.01 to 1.39 | 1.18 | 1.01 to 1.39 | 14 352 | 2100 |
| Population 10 001–50 000 | 1.29 | 1.13 to 1.47 | 1.27 | 1.11 to 1.45 | 16 583 | 2443 |
| Population >50 000 | 1.29 | 1.12 to 1.49 | 1.28 | 1.10 to 1.48 | 4259 | 639 |

*Model adjusted for monthly per capita income (BR$), Psychosocial Care Centre coverage, urbanisation, percentage of people unemployed, percentage of individuals who were divorced, percentage of Pentecostal Christians and percentage of households with one resident.
†Change in suicide rate per 100% increase in homicide rate.
‡This number excludes municipalities with zero suicides recorded between 2008 and 2014.
RR, rate ratio.

**Table 3** Unadjusted and adjusted associations between suicide rates and homicide rates in the Brazilian municipalities by quintiles of homicide rates, 2008–2014*

| Homicide rate | Unadjusted model | | Adjusted model† | |
|---|---|---|---|---|
| | RR‡ | 95% CI | RR‡ | 95% CI |
| First quintile (reference) | 1 | | 1 | |
| Second quintile | 1.01 | 0.96 to 1.06 | 1.01 | 0.96 to 1.06 |
| Third quintile | 1.04 | 0.99 to 1.09 | 1.04 | 0.99 to 1.09 |
| Fourth quintile | 1.10 | 1.04 to 1.15 | 1.09 | 1.04 to 1.14 |
| Fifth quintile | 1.13 | 1.07 to 1.19 | 1.12 | 1.06 to 1.18 |

*Based on 35 357 suicides from 5051 municipalities. Municipalities with zero suicides recorded between 2008 and 2014 were excluded.
†Model adjusted for monthly per capita income (BR$), Psychosocial Care Centre coverage, urbanisation, percentage of unemployed people, percentage of individuals who were divorced, percentage of Pentecostal Christians and percentage of households with one resident.
‡Change in suicide rate per 100% increase in homicide rate.
RR, rate ratio.

to be robust, with minimal changes in magnitude between models accounting for different potential sources of bias.

Some limitations of this study are acknowledged. This study used an ecological design, and therefore conclusions are applicable to populations rather than individuals. An ecological design was used due to data availability, and only the use of aggregate data allows the study of the total Brazilian population longitudinally. Furthermore, given our interest in homicide rates as an indicator of contextual violence, aggregate data for homicide were required.

Another limitation could be that large municipalities and small municipalities have equal weight in the analysis. For this reason, we have also provided estimates after stratification by municipal population size.

Potential bias resulting from misclassification of suicides and homicides is another possible limitation. Homicide and suicide deaths may have a risk of misclassification as events of undetermined intent or unspecified accidents. Suicide may be under-reported due to stigmatisation and social taboo, while homicides could be under-reported as

they may indicate unrest or instability. It is unclear whether the frequency of misclassification differs between homicides and suicides. If misclassification of homicides and suicides occurred equally frequently, then the observed estimates of association between homicide and suicide rates are conservative. In fact, the observed estimate of association was slightly stronger when only municipalities judged to have high-quality vital information were included. Nevertheless, misclassification is not expected to be a major concern because all deaths due to reported external causes, including homicides and suicides, are determined based on autopsy.[19]

### Results in context
The results of this study are in contrast with a previous cross-sectional ecological study of the association between homicide and suicide rates in Brazil, which observed a significant inverse correlation between the two forms of death. In their study, Bando and Lester[16] found a correlation of −0.61 (p<0.001) between age-standardised

**Table 4** Unadjusted and adjusted associations between suicide rates and homicide rates in the Brazilian municipalities, by sex and age groups, 2008–2014

| Age range | Sex | Unadjusted model | | Adjusted model* | | Number of suicides | Number of municipalities‡ |
|---|---|---|---|---|---|---|---|
| | | RR† | 95% CI | RR† | 95% CI | | |
| 10–19 | Men | 1.24 | 0.91 to 1.71 | 1.20 | 0.87 to 1.66 | 10 626 | 1518 |
| | Women | 1.53 | 0.96 to 2.42 | 1.53 | 0.96 to 2.43 | 6167 | 881 |
| 20–59 | Men | 1.29 | 1.17 to 1.43 | 1.24 | 1.12 to 1.38 | 32 382 | 4626 |
| | Women | 1.23 | 1.00 to 1.51 | 1.20 | 0.98 to 1.47 | 19 187 | 2741 |
| 60 and older | Men | 1.12 | 0.90 to 1.39 | 1.05 | 0.84 to 1.32 | 19 096 | 2728 |
| | Women | 1.39 | 0.89 to 2.18 | 1.28 | 0.80 to 2.01 | 7455 | 1065 |

*Model adjusted for monthly per capita income (BR$), Psychosocial Care Centre coverage, urbanisation, percentage of unemployed people, percentage of individuals who were divorced, percentage of Pentecostal Christians and percentage of households with one resident.
†Change in suicide rate per 100% increase in homicide rate.
‡This number excludes municipalities with zero suicides recorded between 2008 and 2014.
RR, rate ratio.

homicide rates and age-standardised suicide rates in 27 Brazilian states, and after stratification by sex, the correlation remained significant in men but not women. Our unadjusted analysis using only the 2010 data showed similar results to those observed by Bando and Lester, however, after adjusting for confounders the association was no longer significant. Confounding factors appeared to have considerable influence on the observed relationship, and the inverse association in the unadjusted analysis is likely a reflection of bias. Further, it is notable that data for our analysis were aggregated to the municipality-level (n=5507) rather than state-level (n=27). Using more fine-grained geographical areas likely resulted in less heterogeneity within regions in terms of important determinants of homicide and suicide.

Our study found some differences in the association by age group. We found no association between increased homicide rates with increased suicide among those 10 to 19 years old or 60 years and older. However, a statistically significant association was observed among men and women 20–59 years old. It is possible that the limited number of suicides among the youngest and oldest age–sex stratifications resulted in insufficient power to achieve significance. Nevertheless, this observation is consistent with findings from a study of the correlations between homicide and suicide rates in Canada and USA, which also showed the strongest correlations among 35–44 and 45–54 year-olds.[28] Interestingly, there was nearly no difference in the strength of association between men and women within this age group, while both suicide and homicide rates tend to be higher in men.[4 6] This finding aligns with other evidence that homicide and suicide rates in men and women tend to fluctuate similarly over time, despite differences in absolute rates, which may be because many predisposing factors for homicide and suicide are similar in men and women.[29]

The relationship between homicide and suicide in the population has been studied for centuries and several theoretical frameworks for understanding their relationship have been suggested.[13] Historically, homicide and suicide have been conceptualised as alternate expressions of the same underlying processes, therefore suggesting an inverse association between the two forms of death. For example, Henry and Short[30] viewed them as alternate expressions of aggression. More recently, Unnithan and colleagues[31] developed the *stream analogy* that homicide and suicide are alternate forms of violence that depend on factors that indicate the amount of violence (forces of production) and whether the violence is expressed as homicide or suicide (forces of direction). On the contrary, several others have questioned the inverse relationship, primarily arising from studies showing similar trends in homicide and suicide rates over time.[32 33] For example, Holinger and Klemen[33] studied mortality rates between 1900 and 1975 and found that rates of homicide, suicide and accidental death tended to fluctuate similarly over time. McKenna, Kelleher and Corcoran (1997) also believed the relationship between homicide and suicide

rates to be positive, reflecting the degree of social disorder in a population. Using data from the Republic of Ireland and Northern Ireland between 1950 and 1990, they observed a positive correlation during times of peace, but an inverse correlation during a war-like period.[34] Lester[35] also questioned whether suicide and homicide are opposites based on findings from a literature review despite interpreting earlier evidence in favour of Henry and Short's hypothesis.[36]

The wide range of theories about the relationship reflects the highly contradictory findings of existing research. Studies based on multi-national data suggest differential relationships by country or geographical area. For example, using cross-sectional data from a large sample of UN member states for 1990–1999 compiled by WHO, Bills and Li[14] examined the correlation between homicide and suicide rates within each region. The strength and direction of the relationship depended heavily on geographical region; a positive correlation between homicide and suicide rates was observed for Europe (r=0.89), while an inverse correlation was found for the Americas (r=−06.2) and Asian Pacific region (−0.97). However, these results were not replicated in a later study by Fountoulakis and Gonda[15], which used data from several countries between 2000 and 2010 that was also compiled by WHO.

Importantly, many of the existing studies have examined the association with limited control for potential confounding factors, which may help to explain the variations in correlation estimates observed between studies. Large areas, such as whole countries, likely contain considerable variability in the distributions of these important demographic and socioeconomic factors that influence homicide and suicide rates and estimates of their association. Furthermore, few studies have accounted for temporality, which may have an important influence on the association. Therefore, better evidence, accounting for confounding factors, is required in order to draw clearer conclusions about the association beyond Brazil.

We observed a positive relationship between homicide and suicide rates that persisted even after adjusting for several important factors including per capita income, CAPS coverage, urbanisation, unemployment, single-person households, divorce and religious affiliation. Therefore, we have evidence that, in Brazil, the relationship exists, even after accounting for these factors.

A positive relationship between homicide and suicide rates suggests that they may share some common social causes, yet no single theory appears to adequately explain the mechanism underlying this relationship. Statistically, a relationship may arise due to a causal association between homicide and suicide rates or due to shared causal factors. For example, high homicide rates may cause high suicide rates indirectly through low social cohesion within the geographical area under study. Alternatively, low social cohesion may cause high homicide rates and high suicide rates independently, therefore producing a spurious relationship. Two covariates we considered—single-person

households and the percentage of individuals within the municipality that are divorced—may contribute to social cohesion. Since the relationship persists after adjusting for these, it is unlikely that low social cohesion adequately explains the relationship between homicide and suicide rates within the Brazilian municipalities. Notably, low social cohesion is only one possible explanation and there may be many others. Taken together, this is consistent with a hypothesis that changes in homicide rates lead to changes in suicide rates, though we cannot be confident that this reflects a causal association. The relationship is likely complex, involving the interaction of social, cultural, economic and political factors.

Targets of homicide and suicide prevention often have highly divergent orientations; while homicide prevention often involves the criminal justice system, suicide prevention usually involves the mental health system and reducing availability of methods for suicide. In Brazil, suicide is expected to be prevented at the community level of care. After the Psychiatric Reformation Movement, mental healthcare in Brazil was officially reformulated in 1978, and the CAPS were created to provide mental healthcare in the community.[37 38] However, increased CAPS coverage was not found to reduce total suicides in Brazilian municipalities between 2008 and 2012.[22] Likewise, to try to control violence the Brazilian justice system has increased incarceration, and as a result, Brazil has the fourth largest prison population in the world,[6] and homicide rates have increased between 2004 and 2012.[21] This highlights the importance of investigating new factors and potential targets for intervention and prevention of suicide and homicide rates in the country.

The presence of a relationship between the two suggests co-benefits of interventions to reduce homicide and suicide deaths. For example, studies in Brazil have shown an association between high coverage of the Brazilian cash transfer programme (a social welfare programme, which provides financial aid to low income Brazilians) and decreased municipal rate of suicide and homicide.[21 22] In addition, extensive evidence from the USA shows that restricting access to firearms is associated with a reduction in firearm-related homicide and suicide.[39] The correlation between firearm availability and suicide and homicide appears to extend to other countries.[40] In Brazil, a nationwide anti-firearm legislation was passed in 2004. There is some evidence that this legislation reduced both firearm-related suicide and firearm-related homicide, although the evidence for a reduction in total homicide and suicide is less clear.[41 42] Unfortunately, the disarmament law was repealed in 2019. Future research may seek to identify other interventions at the policy level, which may help to reduce both homicide and suicide rates in the population.

## CONCLUSION

Our longitudinal ecological study showed a robust positive relationship between homicide and suicide rates in Brazilian municipalities. We showed a dose–response relationship such that areas of higher homicide rates showed greater increase in suicide rates. The results of this study contribute clear evidence for the relationship to a literature that has previously shown conflicting evidence. A positive relationship between homicide and suicide rates can have important implications for public health and policy interventions as approaches to reduce one of homicide or suicide rates may also help to lower rates of the other. This can have important implications for Brazil and other countries where levels of violence are considerable.

**Author affiliations**
[1]Center of Data and Knowledge Integration for Health, Salvador, Brazil
[2]Centre for Global Mental Health, London School of Hygiene & Tropical Medicine, London, United Kingdom
[3]Division of Psychiatry, University College London, London, UK
[4]Department of Psychiatry, The University of Melbourne, Heidelberg, Victoria, Australia
[5]Institute for Global Health, University College London, London, UK
[6]Instituto de Saúde Coletiva, Federal University of Bahia, Salvador, Bahia, Brazil

**Contributors** DBM, LA, FA and MLB were responsible for the conception of the study. DBM obtained the data and conducted the data analysis. DBM and KM contributed equally to the preparation of the manuscript. DBM, KM, LA, DD, FA, LK, GL and MLB contributed to the interpretation of results, provided critical review of the manuscript and had final responsibility for the decision to submit for publication.

**Funding** The study was in part funded by the Medical Research Council (MC_PC_MR/R018677/1) and Wellcome Trust (202912/Z/16/Z). This work was also supported by the National Institute for Health Research, University College London Hospital, Biomedical Research Centre.

**Disclaimer** The funders were not involved in the design and conduct of the study; collection, management, analysis and interpretation of the data; preparation, review or approval of the manuscript; and decision to submit the manuscript for publication.

**Competing interests** None declared.

**Patient consent for publication** Not required.

**Provenance and peer review** Not commissioned; externally peer reviewed.

**Data availability statement** Data are available in a public, open access repository. All data used in this study are publicly available online (http://www2.datasus.gov.br).

**ORCID iD**
Keltie McDonald http://orcid.org/0000-0002-0204-9049

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
