## [Reviewer comments · BMJ Open]

ARTICLE DETAILS

TITLE (PROVISIONAL)	The association between homicide rates and suicide rates: a countrywide longitudinal analysis of 5,507 Brazilian municipalities
AUTHORS	Machado, Daiane Borges; McDonald, Keltie; Araújo, Luis; Devakumar, Delan; Alves, Flávia; Kiss, Lígia; Lewis, Glyn; Barreto, Mauricio

VERSION 1 – REVIEW

REVIEWER	Melissa Raven University of Adelaide, Australia
REVIEW RETURNED	25-Jun-2020

GENERAL COMMENTS	This is an impressive and important paper. It is well-structured and well-written, and it has been gratifying to review. I have a few minor issues with it. Abstract: If possible, I would like the Conclusions section to be expanded a bit, to say that the results are consistent with a hypothesis that changes in homicide rates lead to changes in suicide rates, but it is not possible to conclude that this reflects a causal association (as discussed on page 22), and that suicide rates have increased despite increased CAPS coverage and homicide rates have increased despite increased incarceration. (as discussed on page 23). Page 8 line 22: What is the rationale for inclusion of percentage of individuals who declared being Pentecostal? I can see the potential relevance of religion, but why specifically Pentecostalism and not also Catholicism, Protestantism etc.? Page 19 line 8: I suggest starting a new paragraph at 'Some limitations...'. Page 23 lines 47-52: I would like to see some suggestions about other interventions at the policy level that might reduce homicide and suicide rates – perhaps funding and support for community development initiatives? Typos page 11, lines 38-40: 'age-standard' should be 'age-standardised' page 14 Table 1: 'Pentecostals' should be 'Pentecostal' page 18 lines 10-12: 'was longer significant' should be 'was no longer significant' page 22 line 40: 'consistent with a hypothesis that homicide rates' should be 'consistent with a hypothesis that changes in homicide rates' page 23 line 49: delete the hyphen in 'policy-level' and delete the comma after 'homicide' page 32 Figure 1 title: 'Homide' should be 'Homicide'
--

REVIEWER	Adalberto Campo-Arias University of Magdalena
-----------------	--

REVIEW RETURNED	14-Aug-2020
-----------------	-------------

GENERAL COMMENTS	It is an original article in the analysis and helps the broader understanding of suicidal behaviours. However,  1. It would be interesting to see more specific details about the relationship between homicide and suicide since there are contradictory findings. 2. Authors need to explicitly present the theories that explain direct association and inverse association. 3. Henry and Short (1954) and Lester (1984) postulated an inverse relationship between homicide and suicide rates. 4. How to explain the difference by gender, by age group and number of inhabitants. 5. Figure 1 is unnecessary—possibly most useful with a statistical test that associates the rates. 6. Reviewing the following references may help to discuss the theories about the association and the comparison with other similar studies: Fountoulakis, K. N., & Gonda, X. (2018). Ancestry and different rates of suicide and homicide in European countries: A study with population-level data. Journal of Affective Disorders, 232, 152-162. Leenen, I., & Cervantes-Trejo, A. (2014). Temporal and geographic trends in homicide and suicide rates in Mexico, from 1998 through 2012. Aggression & Violent Behaviors; 19 (6), 699-707. Lester, D. (1984). The association between the quality of life and suicide and homicide rates. Journal of Social Psychology, 124 (2), 247-248. Lester, D. (1987). Murders and suicide: Are they polar opposites. Behavioral Sciences & the Law, 5 (1), 49-60. Lester, D. (1988). A regional analysis of suicide and homicide rates in the USA: search for broad cultural patterns. Social Psychiatry and Psychiatric Epidemiology, 23 (3), 202-205. Lester, D. (1993). The effect of war on suicide rates. A study of France 1826 to 1913. European Archives of Psychiatry and Clinical Neuroscience, 242 (4): 248-249. Henry, A. F., & Short, J. F. (1954). Suicide and homicide: Some economic, sociological and psychological aspects of aggression (Vol. 91442). New York: Free Press. Wu, B. (2003). Testing the stream analogy for lethal violence: A macro study of suicide and homicide. West Criminology Review, 4 (3), 215-225.
--

VERSION 1 – AUTHOR RESPONSE

Reviewer: 1

Melissa Raven, University of Adelaide, Australia

1. **Abstract: If possible, I would like the Conclusions section to be expanded a bit, to say that the results are consistent with a hypothesis that changes in homicide rates lead to changes in suicide rates, but it is not possible to conclude that this reflects a causal association (as discussed on page 22), and that suicide rates have increased despite increased CAPS coverage and homicide rates have increased despite increased incarceration. (as discussed on page 23).**

Authors' response: This is a helpful suggestion to strengthen the abstract. We have added a sentence to expand the Conclusions section, as follow: "Our results are in line with the hypothesis that changes in homicide rates lead to changes in suicide rates, although a causal association cannot be established from this study. Suicide and homicide rates have increased in Brazil despite increased community mental health support and incarceration, respectively, therefore new avenues for intervention are needed". (page 2-3, lines 50-54).

2. Page 8 line 22: What is the rationale for inclusion of percentage of individuals who declared being Pentecostal? I can see the potential relevance of religion, but why specifically Pentecostalism and not also Catholicism, Protestantism etc.?

Authors' response: We added a sentence with rationale for including percentage of individual who declared being Pentecostal as a confounder: "We included the percentage of individuals who declared being Pentecostal was chosen as a confounder, as in previous studies (1,2), because it is likely to be the best proxy for measuring religious engagement in Brazil; Brazilians who identify themselves as Pentecostal tend to actively attend the church and its activities while those who identify themselves as Catholic seem to do so only referring to their family background, even if they do not participate in any religious activity (3)" (page 10, lines 192-197).

3. Page 19 line 8: I suggest starting a new paragraph at 'Some limitations...'

Authors' response: We added an extra space to clearly show the new paragraph, as suggested (page 20, line 28).

4. Page 23 lines 47-52: I would like to see some suggestions about other interventions at the policy level that might reduce homicide and suicide rates – perhaps funding and support for community development initiatives?

Authors' response: We added information about other intervention that seem to be associated with decreased suicide and homicide rates in Brazil, as follow: "studies in Brazil have shown an association between high coverage of the Brazilian cash transfer program (a social welfare program, which provides financial aid to low income Brazilians) and decreased municipal rate of suicide and homicide (2,4)" (page 26, lines 193-196).

5. Typos

- page 11, lines 38-40: 'age-standard' should be 'age-standardised'
- page 14 Table 1: 'Pentecostals' should be 'Pentecostal'
- page 18 lines 10-12: 'was longer significant' should be 'was no longer significant'
- page 22 line 40: 'consistent with a hypothesis that homicide rates' should be 'consistent with a hypothesis that changes in homicide rates'
- page 23 line 49: delete the hyphen in 'policy-level' and delete the comma after 'homicide'
- page 32 Figure 1 title: 'Homide' should be 'Homicide'

Authors' response: We corrected all of the typos listed above.

VERSION 2 – REVIEW

REVIEWER	Melissa Raven University of Adelaide, Australia
REVIEW RETURNED	06-Oct-2020
GENERAL COMMENTS	I am happy with the changes. The paper was already good, but it is better now. Thank-you for the clarification about religion. (I do not think

	Pentecostalism would be the best proxy in my country, but I may be wrong.)
REVIEWER	Adalberto Campo-Arias University of Magdalena, Colombia.
REVIEW RETURNED	22-Sep-2020
GENERAL COMMENTS	The new version summarizes much of the suggestions with an extensive review of divergent theories about the relationship between homicide and suicide rates. No additional modifications are necessary.